# DOMAIN AGGREGATION NETWORKS FOR MULTI-SOURCE DOMAIN ADAPTATION

## ABSTRACT

In many real-world applications, we want to exploit multiple source datasets of similar tasks to learn a model for a different but related target dataset – e.g., recognizing characters of a new font using a set of different fonts. While most recent research has considered ad-hoc combination rules to address this problem, we extend previous work on domain discrepancy minimization to develop a finite-sample generalization bound, and accordingly propose a theoretically justified optimization procedure. The algorithm we develop, Domain AggRegation Network (DARN), is able to effectively adjust the weight of each source domain during training to ensure relevant domains are given more importance for adaptation. We evaluate the proposed method on real-world sentiment analysis, digit recognition and object recognition datasets and show that DARN can significantly outperform the state-of-the-art alternatives.

## 1 INTRODUCTION

Many machine learning algorithms assume the learned predictor will be tested on the data from the same distribution as the training data. This assumption, although reasonable, is not necessarily true for many real-world applications. For example, patients from one hospital may have a different distribution of gender, height and weight from another hospital. Consequently, a diagnostic model constructed at one location may not be directly applicable to another location without proper adjustment. The situation becomes even more challenging when we want to use data from multiple *source domains* to build a model for a *target domain*, as this requires deciding, e.g., how to rank the source domains and how to effectively aggregate these domains when training complex models like deep neural networks. Including irrelevant or worse, adversarial, data from certain source domains can severely reduce the performance on the target domain, leading to undesired consequences.

To address this problem, researchers have been exploring methods of *transfer learning* (Pan & Yang, 2009) or *domain adaptation* (Mansour et al., 2009a; Cortes et al., 2019), where a model is trained based on labelled source data and unlabelled target data. Most existing works have focused on single-source to single-target ("one-to-one") domain adaptation, using different assumptions such as covariate shift (Shimodaira, 2000; Gretton et al., 2009; Sugiyama & Kawanabe, 2012) or concept drift (Jiang & Zhai, 2007; Gama et al., 2014). When dealing with multiple source domains, one may attempt to directly use these approaches by combining all source data into a large joint dataset and then apply one-to-one adaptation. This naïve aggregation method will often fail as not all source domains are equally important when transferring to a specific target domain. There are some works on multi-source to single-target adaptation. Although many of them are theoretically motivated with cross-domain generalization bounds, they either use ad-hoc aggregation rules when developing actual algorithms (Zhao et al., 2018; Li et al., 2018) or lack finite-sample analysis (Mansour et al., 2009b;c; Hoffman et al., 2018a). This leaves a gap between the theory for multi-source adaptation and theoretically sound algorithm for domain aggregation.

This research has three contributions: First, we extend prior work on one-to-one adaptation using discrepancy (Cortes et al., 2019) to develop a finite-sample cross-domain generalization bound for multi-source adaptation. We show that in order to improve performance on the specific target domain, there is a trade-off between utilizing all source domains to increase effective sample size, versus removing source domains that are underperforming or not similar to the target domain. Second, motivated by our theory and domain adversarial method (Ganin & Lempitsky, 2015; Ganin

et al., 2016), we propose Domain AggRegation Network (DARN), that can effectively aggregate multiple source domains dynamically during the course of training. Unlike previous works, our aggregation scheme (Eq. (6)), which itself is of independent interest in some other contexts, is a direct optimization of our generalization upper bound (Eq. (3)) without resorting to heuristics. Third, our experiments on sentiment analysis, digit recognition and object recognition show that DARN can significantly outperform state-of-the-art methods.

Section 2 introduces necessary background on one-to-one adaptation based on discrepancy. Then Section 3 elaborates our theoretical analysis and the corresponding algorithm deployment. Section 4 discusses about related approaches in more detail, highlighting the key differences to the approach developed here. Section 5 empirically compares the performance of the proposed method to other alternatives. Finally, Section 6 concludes this work.

## 2 BACKGROUND ON CROSS-DOMAIN GENERALIZATION

This section provides necessary background from previous work on one-to-one domain adaptation (Cortes et al., 2019). Let $\mathcal{X}$ be the input space, $\mathcal{Y} \subseteq \mathbb{R}$ be output space and $\mathcal{H} \subseteq \{h : \mathcal{X} \mapsto \mathcal{Y}\}$ be a hypothesis set. A loss function $L : \mathcal{Y} \times \mathcal{Y} \mapsto \mathbb{R}^+$ is $\mu$-admissible[1] if

$$\forall y, y', y'' \in \mathcal{Y} \qquad |L(y', y) - L(y'', y)| \leq \mu |y' - y''|. \tag{1}$$

The *discrepancy* (Mansour et al., 2009a) between two distributions $P, Q$ over $\mathcal{X}$ is defined as

$$\text{disc}(P, Q) = \max_{h, h' \in \mathcal{H}} |\mathcal{L}_P(h, h') - \mathcal{L}_Q(h, h')| \quad \text{where} \quad \mathcal{L}_P(h, h') = \mathbb{E}_{x \sim P}[L(h(x), h'(x))].$$

This quantity can be computed or approximated empirically given samples from both distributions. For classification problems with 0-1 loss, it reduces to the well-known $d_{\mathcal{A}}$-distance (Kifer et al., 2004; Blitzer et al., 2008; Ben-David et al., 2010) and can be approximated using a domain-classifier loss w.r.t. $\mathcal{H}$ (Zhao et al., 2018; Ben-David et al., 2007, Sec.4), while for regression problems with $L_2$ loss, it reduces to a maximum eigenvalue (Cortes & Mohri, 2011, Sec.5); see Section 3.2 for more details. For two domains $(P, f_P), (Q, f_Q)$ where $f_P, f_Q : \mathcal{X} \mapsto \mathcal{Y}$ are the corresponding labeling functions, we have the following cross-domain generalization bound:

**Theorem 1 (Proposition 5 & 8, Cortes et al. (2019))** *Let $\mathfrak{R}_m(\mathcal{H})$ be the Rademacher complexity of $\mathcal{H}$ given sample size $m$, $\mathcal{H}_Q = \{x \mapsto L(h(x), f_Q(x)) : h \in \mathcal{H}\}$ be the set of functions mapping $x$ to its loss w.r.t. $f_Q$ and $\mathcal{H}$,*

$$\eta_{\mathcal{H}} = \mu \times \min_{h \in \mathcal{H}} \left( \max_{x \in \text{supp}(\widehat{P})} |f_P(x) - h(x)| + \max_{x \in \text{supp}(\widehat{Q})} |f_Q(x) - h(x)| \right), \tag{2}$$

*be a constant measuring how well $\mathcal{H}$ can fit the true models where $\text{supp}(\widehat{P})$ means the support of the empirical distribution $\widehat{P}$ (using the $\mu$ from Eq. (1)), and $M_Q = \sup_{x \in \mathcal{X}, h \in \mathcal{H}} L(h(x), f_Q(x))$ be the upper bound on loss for $Q$. Given $\widehat{Q}$ with $m$ points sampled iid from $Q$ labelled according to $f_Q$, for $\delta \in (0, 1), \forall h \in \mathcal{H}$, w.p. at least $1 - \delta$,*

$$\mathcal{L}_P(h, f_P) \leq \mathcal{L}_{\widehat{Q}}(h, f_Q) + \text{disc}(P, Q) + 2\mathfrak{R}_m(\mathcal{H}_Q) + \eta_{\mathcal{H}} + M_Q \sqrt{\frac{\log(1/\delta)}{2m}}.$$

This theorem provides a way to generalize across domains when we have sample $\widehat{Q}$ labelled according to $f_Q$ and an unlabelled sample $\widehat{P}$. The first term is the usual loss function for the sample $\widehat{Q}$, while the second term $\text{disc}(P, Q)$ can be estimated based on the unlabelled data $\widehat{Q}, \widehat{P}$. $\eta_{\mathcal{H}}$ measures how well the model family $\mathcal{H}$ can fit the example from the datasets, and it is not controllable once $\mathcal{H}$ is given. The final term, as a function of sample size $m$, determines the convergence speed.

---

[1]For example, the common $L_p$ losses are $\mu$-admissible (Cortes et al., 2019, Lemma 23).

## 3 DOMAIN AGGREGATION NETWORKS

### 3.1 THEORY

Suppose we are given $k$ source domains $\{(S_i, f_{S_i}) : i \in [k] \overset{\text{def}}{=} \{1, 2, \ldots, k\}\}$ and a target domain $(T, f_T)$ where $S_i, T$ are distributions over $\mathcal{X}$ and $f_{S_i}, f_T : \mathcal{X} \mapsto \mathcal{Y}$ are their respective labelling functions. For simplicity, assume that each sample $\widehat{S}_i$ has $m$ points, drawn iid from $S_i$ and labelled according to $f_{S_i}$. We are also given $m$ unlabelled points $\widehat{T}$ drawn iid from $T$. We want to leverage all source domains' information to learn a model $h \in \mathcal{H}$ minimizing $\mathcal{L}_T(h, f_T)$.

One naïve approach could be to combine all the source domains into a large joint dataset and conduct one-to-one adaptation to the target domain using Theorem 1. However, including data from irrelevant or even adversarial domains is likely to jeopardize the performance on the target domain, which is sometimes referred to as *negative transfer* (Pan & Yang, 2009). Moreover, as certain source domains may be more similar or relevant to the target domain than the others, it makes more sense to adjust their importance according to their utilities. We propose to find domain weight $\alpha_i \geq 0$ such that $\sum_{i=1}^{k} \alpha_i = 1$ to achieve this. Our main theorem below sheds some light on how we should combine the source domains (the proof is provided in Appendix A):

**Theorem 2** *Given $k$ source domains datasets $\{(x_j^{(i)}, y_j^{(i)}) : i \in [k], j \in [m]\}$ with $m$ iid examples each where $\widehat{S}_i = \{x_j^{(i)}\}$ and $y_j^{(i)} = f_{S_i}(x_j^{(i)})$, for any $\boldsymbol{\alpha} \in \Delta = \{\boldsymbol{\alpha} : \alpha_i \geq 0, \sum_i \alpha_i = 1\}, \delta \in (0, 1)$, and $\forall h \in \mathcal{H}$, w.p. at least $1 - \delta$*

$$\mathcal{L}_T(h, f_T) \leq \sum_i \alpha_i \left( \mathcal{L}_{\widehat{S}_i}(h, f_{S_i}) + \mathrm{disc}(T, S_i) + 2\mathfrak{R}_m(\mathcal{H}_{S_i}) + \eta_{\mathcal{H}, i} \right) + \|\boldsymbol{\alpha}\|_2 M_S \sqrt{\frac{\log(1/\delta)}{2m}},$$
(3)

*where $\mathcal{H}_{S_i} = \{x \mapsto L(h(x), f_{S_i}(x)) : h \in \mathcal{H}\}$ is the set of functions mapping $x$ to the corresponding loss, $\eta_{\mathcal{H}, i}$ is a constant similar to Eq. (2) with $\widehat{Q} = \widehat{S}_i$, $\widehat{P} = \widehat{T}$ and $M_S = \sup_{i \in [k], x \in \mathcal{X}, h \in \mathcal{H}} L(h(x), f_{S_i}(x))$ is the upper bound on loss on the source domains.*

There are several observations. (1) In the last term of the bound, $m/\|\boldsymbol{\alpha}\|_2^2$ serves as the effective sample size. If $\boldsymbol{\alpha}$ is uniform (i.e., $[1/k, \ldots, 1/k]^\top$), then the effective sample size is $km$; if $\boldsymbol{\alpha}$ is one-hot, then we effectively only have $m$ points from exactly one domain. (2) $\mathfrak{R}_m(\mathcal{H}_{S_i})$ determines how expressive the model family $\mathcal{H}$ is w.r.t. the source data $S_i$. It can be estimated from samples (Bartlett & Mendelson, 2002, Theorem 11), but the computation is non-trivial for a model family like deep neural networks. The $\eta_{\mathcal{H}, i}$ is uncontrollable once the hypothesis class $\mathcal{H}$ (e.g., neural network architecture) is given. Using a richer $\mathcal{H}$ is not always beneficial. Richer $\mathcal{H}$ can reduce the $\eta_{\mathcal{H}, i}$ and also help us find a better function $h$ with smaller source losses $\mathcal{L}_{\widehat{S}_i}$, but it will increase the $\mathfrak{R}_m(\mathcal{H}_{S_i})$. As removing these two terms does not seem to be empirically significant, we ignore them below for simplicity. (3) Let $g_{h,i} \overset{\text{def}}{=} \mathcal{L}_{\widehat{S}_i}(h, f_{S_i}) + \mathrm{disc}(T, S_i)$. Small $g_{h,i}$ indicates that we can achieve small loss on domain $S_i$, and it is similar to the target domain (i.e., small $\mathrm{disc}(T, S_i)$, estimated from $\widehat{T}, \widehat{S}_i$). We may want to emphasize on $S_i$ by setting $\alpha_i$ close to 1, but this will reduce the effective sample size. Therefore, we have to trade-off between the terms in this bound by choosing a proper $\boldsymbol{\alpha}$. (4) When $S_i$ and $T$ are only partially overlapped, it might be difficult to find a suitable $\alpha_i$. In such cases, we can artificially split the given source domains into smaller datasets (e.g., by using clustering) then apply our method on the finer scale. This strategy requires that the learning algorithm has low computational complexities w.r.t. $k$, the number of source domains. As we will see later in Section 3.3, this is indeed the case for our algorithm.

Before we proceed to develop an algorithm based on the theorem, let us compare Eq. (3) to existing finite-sample bounds. The bound of Blitzer et al. (2008, Theorem 3) is informative when we have access to a small set of *labelled* target examples. In such cases, we can improve our bound by using this small labelled target set as an additional source domain $S_{k+1}$. We can also perform better model selection using such labelled set. The bound of Zhao et al. (2018, Theorem 2) is based on the $d_{\mathcal{A}}$-distance, which is a special case of $\mathrm{disc}$ in our bound. Moreover, our bound (Eq. (3)) use sample-based Rademacher complexity, which is generally tighter than other complexity measures such as VC-dimension (Bartlett & Mendelson, 2002; Koltchinskii et al., 2002; Bousquet et al., 2003).

## 3.2 ALGORITHM

In this section, we illustrate how to develop a practical algorithm based on Theorem 2. Ignoring the constants, we would like to minimize the upper bound of Eq. (3):

$$\min_{h \in \mathcal{H}} \min_{\boldsymbol{\alpha} \in \Delta} \quad U_h(\boldsymbol{\alpha}) = \langle \mathbf{g}_h, \boldsymbol{\alpha} \rangle + \tau \|\boldsymbol{\alpha}\|_2 \tag{4}$$

where $\mathbf{g}_h = [g_{h,1}, \ldots, g_{h,k}]^\top$ and $\tau > 0$ is a hyper-parameter. If we can solve the inner minimization exactly given $h$, then we can treat the optimal $\boldsymbol{\alpha}^*(h)$ as a function of $h$ and solve the outer minimization over $h$ effectively. In the following, we show how to achieve this.

Given $\mathbf{g}_h$, the inner minimization can be reformulated as a second-order cone programming problem, but it has no closed-form solution due to the $\tau \|\boldsymbol{\alpha}\|_2$ term. Consider the following problem ($\mathbf{z} = -\mathbf{g}_h/\tau$ recovers the inner minimization):

$$\min_{\boldsymbol{\alpha} \in \Delta} \quad -\langle \mathbf{z}, \boldsymbol{\alpha} \rangle + \|\boldsymbol{\alpha}\|_2 \tag{5}$$

The Lagrangian for its dual problem is

$$\Lambda(\boldsymbol{\alpha}, \boldsymbol{\lambda}, \nu) = -\mathbf{z}^\top \boldsymbol{\alpha} + \|\boldsymbol{\alpha}\|_2 - \boldsymbol{\lambda}^\top \boldsymbol{\alpha} + \nu(\mathbf{1}^\top \boldsymbol{\alpha} - 1) \qquad \nu \in \mathbb{R}, \boldsymbol{\lambda} \succeq 0$$

Taking the derivative w.r.t. $\boldsymbol{\alpha}$ and setting it to zero gives

$$\frac{\partial \Lambda}{\partial \boldsymbol{\alpha}} = -\mathbf{z} + \boldsymbol{\alpha}/\|\boldsymbol{\alpha}\|_2 - \boldsymbol{\lambda} + \nu \mathbf{1} = 0 \quad \Longrightarrow \quad \boldsymbol{\alpha}^*/\|\boldsymbol{\alpha}^*\|_2 = \mathbf{z} - \nu \mathbf{1} + \boldsymbol{\lambda}.$$

Notice that $\boldsymbol{\alpha} \neq \mathbf{0}$ so we have the constraint $\|\mathbf{z} - \nu\mathbf{1} + \boldsymbol{\lambda}\|_2 = 1$. Using this $\boldsymbol{\alpha}^*$ in $\Lambda$ gives the following dual problem

$$\max_{\nu, \boldsymbol{\lambda}} \quad -\nu \qquad \text{s.t.} \quad \|\mathbf{z} - \nu\mathbf{1} + \boldsymbol{\lambda}\|_2 = 1 \quad \text{and} \quad \boldsymbol{\lambda} \succeq 0$$

We would like to *decrease* $\nu$ as much as possible, and the best we can attain is the $\nu^*$ satisfying $\|[\mathbf{z} - \nu^*\mathbf{1}]_+\|_2 = 1$ where $[\mathbf{v}]_+ = \max(\mathbf{0}, \mathbf{v})$. In this case, the optimal $\boldsymbol{\lambda}^*$ can be attained as $\lambda_i^* = 0$ if $z_i - \nu^* > 0$, otherwise $\lambda_i^* = \nu^* - z_i$; see Fig. 1. Although we do not have a closed-form expression for the optimal $\nu^*$, we can use binary search to find it, starting from the interval $[z_{\min} - 1, z_{\max}]$ where $z_{\min}, z_{\max}$ are the minimum and maximum of $\mathbf{z}$ respectively. Then we can recover the primal solution as

$$\boldsymbol{\alpha}^* = [\mathbf{z} - \nu^*\mathbf{1}]_+ / \|[\mathbf{z} - \nu^*\mathbf{1}]_+\|_1. \tag{6}$$

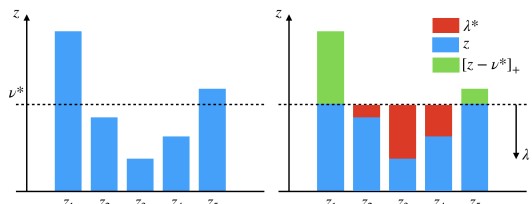

Figure 1: Optimal solution to Eq. (5) (best viewed in color). $\boldsymbol{\lambda}^*$ compensates what is below the $\nu^*$, while $\nu^*$ is chosen such that the vector of the green bars has $L_2$ norm of 1.

Eq. (6) gives rise to a new way to project any vector $\mathbf{z}$ to the probability simplex, which is of independent interest and may be used in some other contexts.[2] It resembles the standard projection onto the simplex based on *squared* Euclidean distance (Duchi et al., 2008, Eq.(3)). One subtle but crucial difference is that our Eq. (5) uses $\|\boldsymbol{\alpha}\|_2$ instead of $\|\boldsymbol{\alpha}\|_2^2$. Recall that $\mathbf{z} = -\mathbf{g}_h/\tau$. Here $\tau$ can be interpreted as a temperature parameter. On one hand, if $\tau \gg 0$, all $\mathbf{z}$ will have similar values and thus the optimal $\nu^*$ will be close to $z_{\max}$ and $\boldsymbol{\alpha}^*$ will be close to uniform. On the other hand, as $\tau \to 0$, $z_{\max}$ will stand out from the rest $z_i$ and eventually $\nu^* = z_{\max} - 1$. This means the $g_{h,i}$ corresponding to the $z_{\max}$ is small enough so we focus solely on this domain and ignore all other domains (even though this will reduce effective sample size as discussed in Section 3.1).

From a different perspective, if we define $F_\Delta^*(\boldsymbol{\alpha}) = \|\boldsymbol{\alpha}\|_2$, then Eq. (5) equals $-F_\Delta(\mathbf{z})$, where $F_\Delta, F_\Delta^*$ are convex conjugates of each other, where we use $\Delta$ to emphasize their dependency on the simplex domain. Then our objective Eq. (4) can be expressed as

$$\min_{h \in \mathcal{H}} \quad -F_\Delta(\mathbf{z}) = -F_\Delta(-\mathbf{g}_h/\tau)$$

---

[2]For example, if we consider $\mathbf{z}$ to be the logits of the final classification layer of a neural network, this projection provides another way to produce class probabilities similar to the softmax transformation.

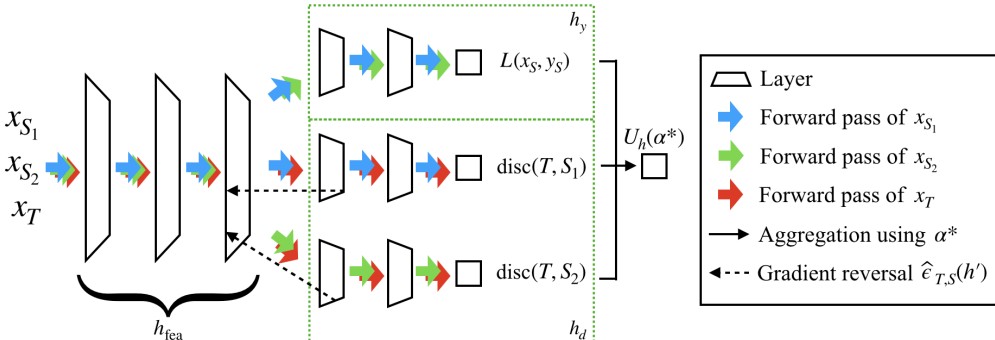

Figure 2: Model architecture with two source domains (best viewed in color). Mini-batches of $x_{S_i}$ and $x_T$ are fed to the network. $x_{S_i}$ will go through the classification/regression path on the upper $h_y$ box, while all $x$ will go through to the corresponding discrepancies (in the lower $h_d$ box). The gradients from the discrepancies will be reverted during backpropagation.

We can optimize our objective and train a neural network $h$ using gradient-based optimizer. The optimal $\boldsymbol{\alpha}^*(h)$ is merely a function of $h$ and we can backprop through it. To facilitate the gradient computation and show that we can efficiently backprop through the projection of Eq. (6), we derive the Jacobian $J = \partial\boldsymbol{\alpha}/\partial\mathbf{z}$ in Appendix B. This ensures effective end-to-end training.

Now we elaborate more on how to compute $g_{h,i}$. The $\mathcal{L}_{\widehat{S}_i}(h, f_{S_i})$ is the task loss. The $\mathrm{disc}(T, S_i)$ depends on whether the task is classification or regression. For classification, disc coincides with the $d_{\mathcal{A}}$-distance, so we use the domain classification loss (Zhao et al., 2018; Ben-David et al., 2007):

$$\mathrm{disc}(\widehat{T}, \widehat{S}_i) = 2\left(1 - \min_{h_d}\widehat{\epsilon}_{T,S_i}(h_d)\right) \qquad \widehat{\epsilon}_{T,S_i}(h_d) = \frac{1}{2m}\sum_{i=1}^{2m}\left|h_d(x) - \delta_{x\in\widehat{T}}\right|$$

where $\widehat{\epsilon}_{T,S_i}(h_d)$ is the sample domain classification loss of a domain classifier $h_d : \mathcal{X} \mapsto \{0, 1\}$. This minimization over $h_d$ will become maximization once we move it outside of the disc due to the minus sign. Then our objective consists of $\min_h$ and $\max_{h_d}$, which resembles adversarial training (Goodfellow et al., 2014): learning a task classifier $h$ to minimize loss and a domain classifier $h_d$ to maximize domain confusion. More specifically, if we decompose the neural network $h$ into a feature extractor $h_{\mathrm{fea}}$ and a label predictor $h_y$ (i.e., $h(x) = h_y(h_{\mathrm{fea}}(x))$), we can learn a domain-classifier $h_d$ on top of $h_{\mathrm{fea}}$ to classify $h_{\mathrm{fea}}(x)$ between $S_i$ and $T$ as a binary classification problem, where the domain itself is the label (see Fig. 2). To achieve this, we use the logistic loss to approximate $\widehat{\epsilon}_{T,S_i}(h_d)$ and apply the gradient reversal layer (Ganin & Lempitsky, 2015; Ganin et al., 2016) when optimizing disc through backpropagation.

If we are solving regression problems with $L_2$ loss, then $\mathrm{disc}(\widehat{T}, \widehat{S}_i) = \|M_T - M_{S_i}\|_2$ is the largest eigenvalue (in magnitude) of the difference of two matrices $M_T = \frac{1}{m}\sum_j h_{\mathrm{fea}}(x_j^{(T)})h_{\mathrm{fea}}^\top(x_j^{(T)})$ and $M_{S_i} = \frac{1}{m}\sum_j h_{\mathrm{fea}}(x_j^{(i)})h_{\mathrm{fea}}^\top(x_j^{(i)})$ (Mansour et al., 2009a; Cortes & Mohri, 2011, Sec.5), which can be conveniently approximated using mini-batches and a few steps of power iteration.

### 3.3 COMPLEXITY

Here we analyze the time and space complexities of the algorithm in each iteration. Similar to MDAN (Zhao et al., 2018), in each gradient step, we need to compute the task loss $\mathcal{L}_{\widehat{S}_i}(h, f_{S_i})$ and the $\mathrm{disc}(T, S_i)$ (or $d_{\mathcal{A}}$-distance) using mini-batches from each source domain $i \in [k]$. The question is whether one can maintain the $O(k)$ complexity given that we need to compute the weights using Eq. (6) and backprop through it. For the forward computation of $\boldsymbol{\alpha}^*$, in order to compute the threshold $\nu^*$ to the $\epsilon > 0$ relative precision, the binary search will cost $O(k\log(1/\epsilon))$. As for the backward pass of gradient computation, according to our calculation in Appendix B, the Jacobian $J = \partial\boldsymbol{\alpha}/\partial\mathbf{z}$ has a concise form, meaning that it is possible to compute the matrix-vector product $J\mathbf{v}$ for a given vector $\mathbf{v}$ in $O(k)$ time and space. Therefore, our space complexity is the same as MDAN but our time complexity is slightly slower by a factor of $\log(1/\epsilon)$. In comparison, the time

complexity for MDMN (Li et al., 2018) is $O(k^2)$ because it requires computing the pairwise weights within the $k$ source domains. When we have a lot of source domains, MDMN will be noticeably slower than MDAN and our DARN.

## 4 RELATED WORK

The idea of utilizing data from the source domain $(S, f_S)$ to train a model for a different but related target domain $(T, f_T)$ has been explored extensively for the last decade using different assumptions (Pan & Yang, 2009; Zhang et al., 2015). For instance, the covariate shift scenario (Shimodaira, 2000; Gretton et al., 2009; Sugiyama & Kawanabe, 2012; Wen et al., 2014) assumes $S \neq T$ but $f_S = f_T$, while concept drift (Jiang & Zhai, 2007; Gama et al., 2014) assumes $S = T$ but $f_S \neq f_T$. More specifically, Theorem 1 (Cortes et al., 2019) shows that both covariate shift (as measured by the discrepancy disc) and model misspecification (as controlled by $\eta_{\mathcal{H}}$) contribute to the adaptation performance (Wen et al., 2014).

Finding a domain-invariant feature space by minimizing a distance measure is common practice in domain adaptation, especially for training neural networks. Tzeng et al. (2017) provided a comprehensive framework that subsumes several prior efforts on learning shared representations across domains (Tzeng et al., 2015; Ganin et al., 2016). DARN uses adversarial domain classifier and the gradient reversal trick from Ganin et al. (2016). Instead of proposing a new loss for each pair of the source and target domains, our main contribution is the aggregation technique of computing the mixing coefficients $\boldsymbol{\alpha}$, which is derived from theoretical guarantees. When dealing with multiple source domains, our aggregation method can certainly be applied to other forms of discrepancies such as MMD (Gretton et al., 2012; Long et al., 2015; 2016), and other model architectures such as Domain Separation Network (Bousmalis et al., 2016), cycle-consistent model (Hoffman et al., 2018b), class-dependent adversarial domain classifier (Pei et al., 2018) and Known Unknown Discrimination (Schoenauer-Sebag et al., 2019).

Our work focuses on multi-source to single-target adaptation, which has been investigated in the literature. Sun et al. (2011) developed a generalization bound but resorted to heuristic algorithms to adjust distribution shifts. Zhao et al. (2018) proposed a certain ad-hoc scheme for the combination coefficients $\boldsymbol{\alpha}$, which, unlike ours, are not theoretically justified. Multiple Domain Matching Network (MDMN) (Li et al., 2018) computes domain similarities not only between the source and target domains but also within the source domain themselves based on Wasserstein-like measure. Calculating such pairwise weights can be computationally demanding when we have a lot of source domains. Their bound requires additional smooth assumptions on the labelling functions $f_{S_i}, f_T$, and is not a finite-sample bound, as opposed to ours. As for the actual algorithm, they also use ad-hoc coefficients $\boldsymbol{\alpha}$ without theoretical justification. Mansour et al. (2009b;c) consider multi-source adaptation where $T = \sum_i \beta_i S_i$ is a convex mixture of source distributions with some weights $\beta_i$. Our analysis does not require this assumption. Hoffman et al. (2018a) provides similar guarantees with different assumptions, but unlike ours, their bounds are not finite-sample bounds.

## 5 EXPERIMENTS

In this section, we compare DARN with several other baselines and state-of-the-art methods for the popular tasks: sentiment analysis, digit recognition and object recognition. The following methods are compared. (1) The **SRC** (for source) method uses only labelled source data to train the model. It merges all available source examples to form a large dataset to perform training without adaptation. (2) **TAR** (for target) is another baseline that uses only $m$ labelled target data instances. It serves as upper bound of the best we can achieve if we had access to the true label of the target data. (3) Domain Adversarial Neural Network (**DANN**) (Ganin et al., 2016) is similar to our method in that we both use adversarial training objectives. It is not obvious how to adapt DANN to the multi-source setting. Here we follow previous protocol (Zhao et al., 2018) and merge all source data to form a large joint source dataset of $km$ instances for DANN to perform adaptation. (4) Moment Matching for Multi-Source Domain Adaptation (**M3SDA**) (Peng et al., 2019) is a recent state-of-the-art method that combines moment matching and maximizing classifier discrepancy (Saito et al., 2018). We use their public code with a few necessary adjustments (change classification head based on the number of classes etc.) (5) Multisource Domain Adversarial Network (**MDAN**) (Zhao et al.,

2018) resembles our method in that we both dynamically assign each source domain an importance weight during training. However, unlike ours, their weights are not theoretically justified. We use the soft version of MDAN since they reported that it performs better than the hard version, and we use their code. (6) Multiple Domain Matching Network (**MDMN**) (Li et al., 2018) computes weights not only between source and target domains but also within source domain themselves. We use their code of computing weights in our implementation. All the methods are applied to the same neural network structure to ensure fair comparison. Our PyTorch implementation will be available online after acceptance.

## 5.1 SENTIMENT ANALYSIS

**Setup**. We use the Amazon review dataset (Blitzer et al., 2007; Chen et al., 2012) that consists of positive and negative product reviews from four domains (Books, DVD, Electronics and Kitchen). Each of them is used in turn as the target domain and the other three are used as source domains. Their sample sizes are 6465, 5586, 7681, 7945 respectively. We follow the common protocol (Chen et al., 2012; Zhao et al., 2018) of using the top-5000 frequent unigrams/bigrams of all reviews as bag-of-words features and train a fully connected model (MLP) with $[1000, 500, 100]$ hidden units for classifying positive versus negative reviews. The dropout drop rate is 0.7 for the input and hidden layers. In each run, we randomly sample 2000 reviews from each domain as labelled source or unlabelled target training examples, while the remaining instances are used as test examples for evaluation. The hyper-parameters are chosen based on cross-validation. The model is trained for 50 epochs and the mini-batch size is 20 per domain. The optimizer is Adadelta with a learning rate of 1.0. The soft version of MDAN has an additional parameter $\gamma = 1/\tau$ which is the inverse of our temperature $\tau$. The chosen parameters are $\gamma = 10.0$ for MDAN and $\gamma = 0.9$ for our DARN, which are selected from a wide range of candidate values.

**Results and Analysis**. Table 1 summarizes the classification accuracies. The last column is the average accuracy of four domains, and the standard errors are calculated based on 20 runs. (1) Most of the time, DANN with joint source data performs worse than SRC which has no adaptation. This may be because it does not adjust for the each source domain, and as a result, it fails to ignore irrelevant data to avoid negative transfer. (2) Some domains are harder to adapt to than the others. For example, the accuracies of SRC and TAR on the Electronics domain are very close to each other, indicating that this requires little to no adaptation. Yet, our DARN is the closest to the TAR performance here. The Books domain is more challenging as the improvement over the SRC method is small, even though there exists a large gap between SRC and TAR. (3) DARN is always within the best performing methods and significantly outperforms others in the Electronics domain. Note that MDMN additionally computes similarities within source domains in each iteration, which can be computationally expensive ($O(k^2)$ per iteration) if the number of source domains is large. Instead, our method focuses on the discrepancy between source and target domains ($O(k)$ per iteration) and can achieve similar performance on this problem.

Table 1: Classification accuracy (%) of the target sentiment datasets. Mean and standard error over 20 runs. The best method(s) (excluding TAR) based on one-sided Wilcoxon signed-rank test at the 5% significance level is(are) shown in bold for each domain.

| Method | Books | DVD | Electronics | Kitchen | Avg. |
|--------|-------|-----|-------------|---------|------|
| SRC | $79.15_{\pm 0.39}$ | $80.38_{\pm 0.30}$ | $85.48_{\pm 0.10}$ | $85.46_{\pm 0.34}$ | $82.62_{\pm 0.20}$ |
| DANN | $79.13_{\pm 0.29}$ | $80.60_{\pm 0.29}$ | $85.27_{\pm 0.14}$ | $85.56_{\pm 0.28}$ | $82.64_{\pm 0.14}$ |
| M3SDA | $79.42_{\pm 0.17}$ | $80.82_{\pm 0.35}$ | $85.52_{\pm 0.19}$ | $86.45_{\pm 0.43}$ | $83.05_{\pm 0.14}$ |
| MDAN | $\mathbf{79.99}_{\pm 0.20}$ | $\mathbf{81.66}_{\pm 0.19}$ | $84.76_{\pm 0.17}$ | $86.82_{\pm 0.13}$ | $83.31_{\pm 0.08}$ |
| MDMN | $\mathbf{80.13}_{\pm 0.20}$ | $\mathbf{81.58}_{\pm 0.21}$ | $85.61_{\pm 0.13}$ | $\mathbf{87.13}_{\pm 0.11}$ | $\mathbf{83.61}_{\pm 0.07}$ |
| DARN | $79.93_{\pm 0.19}$ | $\mathbf{81.57}_{\pm 0.16}$ | $\mathbf{85.75}_{\pm 0.16}$ | $\mathbf{87.15}_{\pm 0.14}$ | $\mathbf{83.60}_{\pm 0.08}$ |
| TAR | $84.10_{\pm 0.13}$ | $83.68_{\pm 0.12}$ | $86.11_{\pm 0.32}$ | $88.72_{\pm 0.14}$ | $85.65_{\pm 0.09}$ |

## 5.2 DIGIT RECOGNITION

**Setup**. Following the setting from previous work (Ganin et al., 2016; Zhao et al., 2018), we use the four digit recognition datasets in this experiment (MNIST, MNIST-M, SVHN and Synth). MNIST

is a well-known gray-scale images for digit recognition, and MINST-M (Ganin & Lempitsky, 2015) is a variant where the black and white pixels are masked with color patches. Street View House Number (SVHN) (Netzer et al., 2011) is a standard digit dataset taken from house numbers in Google Street View images. Synthetic Digits (Synth) (Ganin & Lempitsky, 2015) is a synthetic dataset that mimic SVHN using various transformations. One of the four datasets is chosen as unlabelled target domain in turn and the other three are used as labelled source domains.

MNIST images are resized to $32 \times 32$ and represented as 3-channel color images in order to match the shape of the other three datasets. Each domain has its own given training and test sets when downloaded. Their respective training sample sizes are 60000, 59001, 73257, 479400, and the respective test sample sizes are 10000, 9001, 26032, 9553. In each run, 20000 images are randomly sampled from each domain's training set as actual labelled source or unlabelled target training examples, and 9000 images are randomly sampled from each domain's test set as actual test examples for evaluation. The model structure is shown in Appendix D. There is no dropout and the hyperparameters are chosen based on cross-validation. It is trained for 50 epochs and the mini-batch size is 128 per domain. The optimizer is Adadelta with a learning rate of 1.0. Validation selected $\gamma = 0.5$ for MDAN and $\gamma = 0.1$ for DARN.

**Results and Analysis**. Table 2 shows the classification accuracy of each target dataset over 20 runs. (1) All methods except DANN can consistently outperform SRC. Again, without proper adjustment for each source domain, DANN with joint source data can perform worse than SRC. This suggests the importance of ignoring irrelevant data to avoid negative transfer. (2) DARN significantly outperforms MDAN and MDMN across all four domains, especially on the MNIST-M and SVHN domains. Notice that even though MDAN and MDMN have generalization guarantees, they both resort to ad-hoc aggregation rules to combine the source domains during training. Instead, our aggregation (Eq. (6)) is a direct optimization of the upper bound (Theorem 2) thus is theoretically justified and empirically superior for this problem.

Table 2: Classification accuracy (%) of the target digit datasets. Mean and standard error over 20 runs. The best method (excluding TAR) based on one-sided Wilcoxon signed-rank test at the 5% significance level is shown in bold for each domain.

| Method | MNIST | MNIST-M | SVHN | Synth | Avg. |
|--------|-------|---------|------|-------|------|
| SRC | $96.78 _{\pm 0.08}$ | $60.80 _{\pm 0.21}$ | $68.99 _{\pm 0.69}$ | $84.09 _{\pm 0.27}$ | $77.66 _{\pm 0.14}$ |
| DANN | $96.41 _{\pm 0.13}$ | $60.10 _{\pm 0.27}$ | $70.19 _{\pm 1.30}$ | $83.83 _{\pm 0.25}$ | $77.63 _{\pm 0.35}$ |
| M3SDA | $96.95 _{\pm 0.06}$ | $65.03 _{\pm 0.80}$ | $71.66 _{\pm 1.16}$ | $80.12 _{\pm 0.56}$ | $78.44 _{\pm 0.36}$ |
| MDAN | $97.10 _{\pm 0.10}$ | $64.09 _{\pm 0.31}$ | $77.72 _{\pm 0.60}$ | $85.52 _{\pm 0.19}$ | $81.11 _{\pm 0.21}$ |
| MDMN | $97.15 _{\pm 0.09}$ | $64.34 _{\pm 0.27}$ | $76.43 _{\pm 0.48}$ | $85.80 _{\pm 0.21}$ | $80.93 _{\pm 0.16}$ |
| DARN | $\mathbf{98.09} _{\pm 0.03}$ | $\mathbf{67.06} _{\pm 0.14}$ | $\mathbf{81.58} _{\pm 0.14}$ | $\mathbf{86.79} _{\pm 0.09}$ | $\mathbf{83.38} _{\pm 0.06}$ |
| TAR | $99.02 _{\pm 0.02}$ | $94.66 _{\pm 0.10}$ | $87.40 _{\pm 0.17}$ | $96.90 _{\pm 0.09}$ | $94.49 _{\pm 0.07}$ |

## 5.3 OBJECT RECOGNITION: OFFICE-HOME

To showcase the applicability of our method to more complicated real-world tasks, we use the challenging Office-Home dataset (Venkateswara et al., 2017). It contains images of 65 everyday objects such as spoon, sink, mug and pen from four different domains: Art, Clipart, Product and Real-World. One of the four datasets is chosen as unlabelled target domain in turn and the other three are used as labelled source domains.

The respective sample sizes are 2427, 4365, 4439, 4357. In each run, 2000 images are randomly sampled from each domain as labelled source or unlabelled target training examples, and the rest images are used as test images for evaluation. We use the ResNet50 He et al. (2016) pretrained from the ImageNet in PyTorch as the base network for feature learning and put an MLP with [1000, 500, 100, 65] units on top for classification. It is trained for 50 epochs and the mini-batch size is 32 per domain. The optimizer is Adadelta with a learning rate of 1.0. MDAN uses $\gamma = 1.0$ while DARN uses $\gamma = 0.5$.

Table 3 shows the classification accuracy of each target dataset over 20 runs. Most existing works in the literature focused on single-source to single-target adaptation on this problem (e.g., see (Long et al., 2018)). Compared to them, using multi-source methods can significantly boost performance,

revealing the importance of using multiple source domains when possible. Even though this is a significantly more challenging problem with more classes and much fewer images compared to the digits datasets, our DARN achieves state-of-the-art performance in this setting, excelling existing methods by a noticeable margin.

Table 3: Classification accuracy (%) of the Office-Home datasets. Mean and standard error over 20 runs. The best method (excluding TAR) based on one-sided Wilcoxon signed-rank test at the 5% significance level is shown in bold for each domain.

| Method | Art | Clipart | Product | Real-World | Avg. |
|--------|-----|---------|---------|------------|------|
| SRC | $58.02 _{\pm 0.47}$ | $57.29 _{\pm 0.30}$ | $74.26 _{\pm 0.22}$ | $77.98 _{\pm 0.25}$ | $66.89 _{\pm 0.16}$ |
| DANN | $57.39 _{\pm 0.69}$ | $57.35 _{\pm 0.35}$ | $73.78 _{\pm 0.27}$ | $78.12 _{\pm 0.21}$ | $66.66 _{\pm 0.19}$ |
| M3SDA | $64.05 _{\pm 0.61}$ | $62.79 _{\pm 0.37}$ | $76.21 _{\pm 0.30}$ | $78.63 _{\pm 0.22}$ | $70.42 _{\pm 0.18}$ |
| MDAN | $68.14 _{\pm 0.58}$ | $67.04 _{\pm 0.21}$ | $81.03 _{\pm 0.22}$ | $82.79 _{\pm 0.15}$ | $74.75 _{\pm 0.18}$ |
| MDMN | $68.67 _{\pm 0.55}$ | $67.75 _{\pm 0.20}$ | $81.37 _{\pm 0.18}$ | $83.32 _{\pm 0.14}$ | $75.28 _{\pm 0.15}$ |
| DARN | $\mathbf{70.00} _{\pm 0.38}$ | $\mathbf{68.42} _{\pm 0.14}$ | $\mathbf{82.75} _{\pm 0.21}$ | $\mathbf{83.88} _{\pm 0.16}$ | $\mathbf{76.26} _{\pm 0.13}$ |
| TAR | $71.19 _{\pm 0.38}$ | $79.16 _{\pm 0.16}$ | $90.66 _{\pm 0.15}$ | $85.60 _{\pm 0.14}$ | $81.65 _{\pm 0.12}$ |

## 5.4 VISUALIZING DOMAIN IMPORTANCE

To demonstrate how DARN can effectively aggregate multiple source domains, we visualize the source domain weights (i.e., $\alpha$ in DARN) for the Amazon dataset. We also compared to the weights produced by MDMN, using the original authors' code.

Fig. 3 and Fig. 4 compare the evolution of source domain weights during training. In each subfigure, every row corresponds to the weights of the source domains when learning for one target domain. The white stripe indicates there is no target domain weight and darker stripe means less weight. They are evaluated at the end of each epoch over 50 epochs. To avoid noisy values due to small mini-batch size, the values are exponential moving averages with a decay rate of 0.95. (1) For DARN, as Electronics and Kitchen are more related to each other than Books and DVD, their respective weights remain higher during training. This is reasonable since we have overlapping products (e.g., blenders) in both domains. The $\alpha$ is changing dynamically during training, showing the flexibility of our method to adjust domain importance when needed. (2) In comparison, the domain weights produced by MDMN are not very stable. We take a closer look at the MDMN weights and notice that they change drastically, especially towards the end of the training. It can produce alternating *one-hot* vectors $\alpha$, changing from one domain to a different domain and ignoring the rest. This instability makes their domain weights hard to interpret.

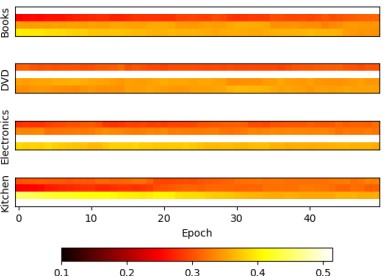

Figure 3: Domain weights of DARN for the Amazon data.

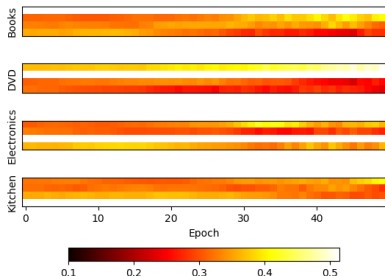

Figure 4: Domain weights of MDMN for the Amazon data.

## 6 CONCLUSION AND FUTURE WORK

Our work uses the discrepancy (Mansour et al., 2009a; Cortes et al., 2019) to derive a finite-sample generalization bound for multi-source to single-target adaptation. Our theorem shows that, in order to achieve the best possible generalization upper bound for a target domain, we need to trade-off between including all source domains to increase effective sample size and removing source domains that are underperforming or not similar to the target domain. Based on this observation, we derive an algorithm, Domain AggRegation Network (DARN), that can dynamically adjust the weight of each source domain during end-to-end training. Experiments on sentiment analysis, digits recognition and object recognition show that DARN outperforms state-of-the-art alternatives. Recent

analysis (Zhao et al., 2019; Johansson et al., 2019) show that solely focusing on learning domain invariant features can be problematic when the marginal label distributions on $\mathcal{Y}$ between source and target domains are significantly different. Thus it makes sense to take $\eta_{\mathcal{H}}$ into consideration when a small amount of labelled data is available for the target domain, which we will explore in the future.

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

## A    PROOF OF THEOREM 2

**Theorem 2** *Given $k$ source domains datasets $\{(x_j^{(i)}, y_j^{(i)}) : i \in [k], j \in [m]\}$ with $m$ iid examples each where $\widehat{S}_i = \{x_j^{(i)}\}$ and $y_j^{(i)} = f_{S_i}(x_j^{(i)})$, for any $\boldsymbol{\alpha} \in \Delta = \{\boldsymbol{\alpha} : \alpha_i \geq 0, \sum_i \alpha_i = 1\}, \delta \in (0, 1)$, and $\forall h \in \mathcal{H}$, w.p. at least $1 - \delta$*

$$\mathcal{L}_T(h, f_T) \leq \sum_i \alpha_i \left( \mathcal{L}_{\widehat{S}_i}(h, f_{S_i}) + \mathrm{disc}(T, S_i) + 2\mathfrak{R}_m(\mathcal{H}_{S_i}) + \eta_{\mathcal{H},i} \right) + \|\boldsymbol{\alpha}\|_2 M_S \sqrt{\frac{\log(1/\delta)}{2m}},$$

*where $\mathcal{H}_{S_i} = \{x \mapsto L(h(x), f_{S_i}(x)) : h \in \mathcal{H}\}$ is the set of functions mapping $x$ to the corresponding loss, $\eta_{\mathcal{H},i}$ is a constant similar to Eq. (2) with $\widehat{Q} = \widehat{S}_i$, $\widehat{P} = \widehat{T}$ and $M_S = \sup_{i \in [k], x \in \mathcal{X}, h \in \mathcal{H}} L(h(x), f_{S_i}(x))$ is the upper bound on loss on the source domains.*

**Proof:** The proof is similar to Cortes et al. (2019); Zhao et al. (2018). Given $\boldsymbol{\alpha} \in \Delta$, the mixture $\widehat{S} = \sum_i \alpha_i \widehat{S}_i$ can be considered as the joint source data with $km$ points, where a point $x^{(i)}$ from $\widehat{S}_i$ has weight $\alpha_i/m$. Define $\Phi = \sup_{h \in \mathcal{H}} \mathcal{L}_T(h, f_T) - \sum_i \alpha_i \mathcal{L}_{\widehat{S}_i}(h, f_{S_i})$. Changing a point $x^{(i)}$ from $\widehat{S}_i$ will change $\Phi$ at most $\frac{M_S \alpha_i}{m}$. Using the McDiarmid's inequality, we have $\Pr(\Phi - \mathbb{E}[\Phi] > \epsilon) \leq \exp\left(-\frac{2\epsilon^2 m}{M_S^2 \|\boldsymbol{\alpha}\|_2^2}\right)$. As a result, for $\delta \in (0, 1)$, w.p. at least $1 - \delta$, the following holds for any $h \in \mathcal{H}$

$$\mathcal{L}_T(h, f_T) \leq \sum_i \alpha_i \mathcal{L}_{\widehat{S}_i}(h, f_{S_i}) + \mathbb{E}[\Phi] + \|\boldsymbol{\alpha}\|_2 M_S \sqrt{\frac{\log(1/\delta)}{2m}}.$$

Now we bound $\mathbb{E}[\Phi]$. Let $\mathcal{H}_{S_i} = \{x \mapsto L(h(x), f_{S_i}(x)) : h \in \mathcal{H}\}$.

$$\mathbb{E}[\Phi] = \mathbb{E}_{\widehat{S}} \left[ \sup_{h \in \mathcal{H}} \mathcal{L}_T(h, f_T) - \sum_i \alpha_i \mathcal{L}_{\widehat{S}_i}(h, f_{S_i}) \right]$$

$$\leq \mathbb{E}_{\widehat{S}} \left[ \sup_{h \in \mathcal{H}} \sum_i \alpha_i \mathcal{L}_{S_i}(h, f_{S_i}) - \sum_i \alpha_i \mathcal{L}_{\widehat{S}_i}(h, f_{S_i}) \right] + \sup_{h \in \mathcal{H}} \left( \mathcal{L}_T(h, f_T) - \sum_i \alpha_i \mathcal{L}_{S_i}(h, f_{S_i}) \right)$$

$$\leq \mathbb{E}_{\widehat{S}} \left[ \sum_i \alpha_i \sup_{h \in \mathcal{H}} \left( \mathcal{L}_{S_i}(h, f_{S_i}) - \mathcal{L}_{\widehat{S}_i}(h, f_{S_i}) \right) \right] + \sum_i \alpha_i \sup_{h \in \mathcal{H}} \left( \mathcal{L}_T(h, f_T) - \mathcal{L}_{S_i}(h, f_{S_i}) \right)$$

$$= \sum_i \alpha_i \mathbb{E}_{\widehat{S}_i} \left[ \sup_{h \in \mathcal{H}} \left( \mathcal{L}_{S_i}(h, f_{S_i}) - \mathcal{L}_{\widehat{S}_i}(h, f_{S_i}) \right) \right] + \sum_i \alpha_i \sup_{h \in \mathcal{H}} \left( \mathcal{L}_T(h, f_T) - \mathcal{L}_{S_i}(h, f_{S_i}) \right)$$

$$\leq 2 \sum_i \alpha_i \mathfrak{R}_m(\mathcal{H}_{S_i}) + \sum_i \alpha_i \sup_{h \in \mathcal{H}} \left( \mathcal{L}_T(h, f_T) - \mathcal{L}_{S_i}(h, f_{S_i}) \right)$$

$$\leq \sum_i \alpha_i \left( 2\mathfrak{R}_m(\mathcal{H}_{S_i}) + \mathrm{disc}(T, S_i) + \eta_{\mathcal{H},i} \right).$$

where first and second inequalities are using the subadditivity of $\sup$, followed by the equality using the independence between the domains $\{\widehat{S}_i\}$, the second last inequality is due to the standard "ghost sample" argument in terms of the Rademacher complexity and the last inequality is due to Cortes et al. (2019, Proposition 8) for each individual $S_i$. ∎

## B  JACOBIAN

Here we calculate the Jacobian $J_{ij} = \partial \alpha_i / \partial z_j$ for Eq. (6) in the main text:

$$\boldsymbol{\alpha}^* = [\mathbf{z} - \nu^*\mathbf{1}]_+ / \|[\mathbf{z} - \nu^*\mathbf{1}]_+\|_1.$$

In the following, we write $\boldsymbol{\alpha} = \boldsymbol{\alpha}^*, \nu = \nu^*$ to simplify notations. Let $S = \{i : z_i - \nu > 0\}$ be the support of the probability vector $\boldsymbol{\alpha}$. $J_{ij} = 0$ if $i \notin S$ or $j \notin S$ since $\alpha_i = 0$ in the former case while $z_j$ does not contribute to the $\boldsymbol{\alpha}$ in the latter case. Now consider the case $i, j \in S$. Let $K = \|[\mathbf{z} - \nu\mathbf{1}]_+\|_1 = \sum_{j \in S}(z_j - \nu)$. Then $\alpha_i = (z_i - \nu) \cdot \frac{1}{K}$ and

$$\frac{\partial \alpha_i}{\partial z_j} = \left(\delta_{i=j} - \frac{\partial \nu}{\partial z_j}\right) \cdot \frac{1}{K} - \frac{1}{K^2} \cdot \frac{\partial K}{\partial z_j} \cdot (z_i - \nu) = \frac{1}{K}\left(\delta_{i=j} - \frac{\partial \nu}{\partial z_j} - \frac{\partial K}{\partial z_j} \cdot \alpha_i\right), \quad (7)$$

where $\delta_{i=j}$ is the indicator or delta function. Now we compute $\frac{\partial \nu}{\partial z_j}$ and $\frac{\partial K}{\partial z_j}$. By the definition of $\nu$, we know that

$$\sum_{j \in S}(z_j - \nu)^2 = |S|\nu^2 - 2\nu \sum_{j \in S} z_j + \sum_{j \in S} z_j^2 = 1$$

$$\implies \quad \nu = \frac{\sum_{j \in S} z_j}{|S|} - \frac{\sqrt{A}}{|S|} \quad \text{where} \quad A = \left(\sum_{j \in S} z_j\right)^2 - |S|\left(\sum_{j \in S} z_j^2 - 1\right)$$

$$\implies \quad \frac{\partial \nu}{\partial z_j} = \frac{1}{|S|} - \frac{B_j}{|S|\sqrt{A}} \quad \text{where} \quad B_j = \sum_{j' \in S} z_{j'} - |S|z_j \quad (8)$$

The first right-arrow is due to the quadratic formula and realizing that $\sum_{j \in S} z_j / |S|$ is the mean of the supported $z_j$ so $\nu$ must be smaller than it (i.e., we take $-$ in the $\pm$ of the quadratic formula, otherwise some of the $z_j$ will not be in the support anymore). And

$$\frac{\partial K}{\partial z_j} = 1 - |S| \cdot \frac{\partial \nu}{\partial z_j} = \frac{B_j}{\sqrt{A}}. \quad (9)$$

Plugging Eq. (8) and Eq. (9) in Eq. (7) gives

$$\frac{\partial \alpha_i}{\partial z_j} = \frac{1}{K}\left[\delta_{i=j} - \frac{1}{|S|} + \frac{B_j}{\sqrt{A}} \cdot \left(\frac{1}{|S|} - \alpha_i\right)\right]$$

Note that

$$\frac{B_j}{K} = \frac{\sum_{j' \in S} z_{j'} - |S|z_j}{\sum_{j' \in S}(z_{j'} - \nu)} = \frac{\sum_{j' \in S}(z_{j'} - \nu) + |S|(\nu - z_j)}{\sum_{j' \in S}(z_{j'} - \nu)} = 1 - |S|\alpha_j.$$

Then

$$J_{ij} = \frac{\partial \alpha_i}{\partial z_j} = \frac{1}{K}\left(\delta_{i=j} - \frac{1}{|S|}\right) + \frac{|S|}{\sqrt{A}}\left(\frac{1}{|S|} - \alpha_i\right)\left(\frac{1}{|S|} - \alpha_j\right).$$

In matrix form,

$$J = \frac{1}{K}\left(\text{Diag}(\mathbf{s}) - \frac{\mathbf{s}\mathbf{s}^\top}{|S|}\right) + \frac{|S|}{\sqrt{A}}\left(\frac{\mathbf{s}}{|S|} - \boldsymbol{\alpha} \circ \mathbf{s}\right)\left(\frac{\mathbf{s}}{|S|} - \boldsymbol{\alpha} \circ \mathbf{s}\right)^\top,$$

where $\mathbf{s} = [s_1, \ldots, s_k]^\top$ is a vector indicating the support $s_i = \delta_{i \in S}$ and $\circ$ is element-wise multiplication. More often, we need to compute its multiplication with a vector $\mathbf{v}$

$$J\mathbf{v} = \frac{\mathbf{s}}{K} \circ \left(\mathbf{v} - \frac{\mathbf{s}^\top \mathbf{v}}{|S|}\right) + \frac{|S|}{\sqrt{A}}\left(\frac{\mathbf{s}}{|S|} - \boldsymbol{\alpha} \circ \mathbf{s}\right)\left(\frac{\mathbf{s}}{|S|} - \boldsymbol{\alpha} \circ \mathbf{s}\right)^\top \mathbf{v}.$$

Note that all quantities except $A$ have been computed during the forward pass of calculating Eq. (6). $A$ can be computed in $O(|S|)$ time so the overall computation is still $O(k)$ since $|S| \leq k$.

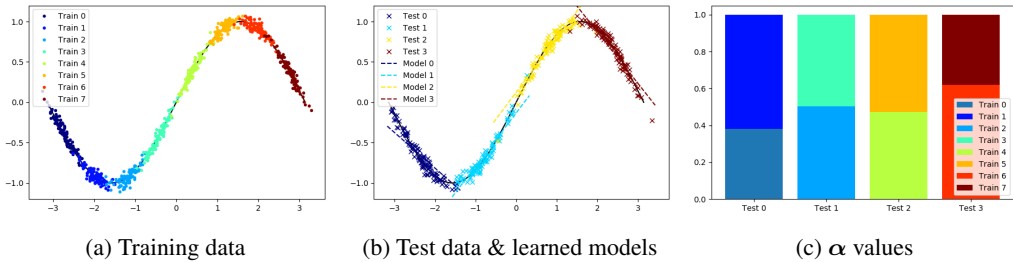

(a) Training data       (b) Test data & learned models       (c) $\boldsymbol{\alpha}$ values

Figure 5: Regression experiment (best viewed in color).

## C  REGRESSION ON SYNTHETIC DATA

Due to the lack of publicly available multi-source regression datasets, we demonstrate our method on a synthetic regression problem here.

**Setup**. We use eight source domains, where the $i$th ($i = \{0, 1, \ldots, 7\}$) domain data is generated by $x_i \sim \mathcal{N}(\frac{\pi}{4}i - \frac{7\pi}{8}, 0.2^2)$ and the output is $y = \sin(x) + \epsilon$ where $\epsilon \sim \mathcal{N}(0, 0.05^2)$ is random noise. These eight domains evenly cover the sin function on $[-\pi, \pi]$ (see Fig. 5a). Next we construct four target domains, where the $j$th ($j = \{0, 1, 2, 3\}$) domain is generated by $x_j \sim \mathcal{N}(\frac{\pi}{2}j - \frac{3\pi}{4}, 0.4^2)$. Similar to the source domains, these four target domains evenly cover $[-\pi, \pi]$ (see Fig. 5b). Each source/target domain has 100 data points.

We use labelled source data and unlabelled target data for learning. We take on one target domain at a time, and learn a linear model from *all* eight source domains with MSE loss. The goal is to see whether our method can focus on the relevant source domains and learn a linear model that can perform well on the target domain.

**Results and Analysis**. First, Fig. 5b shows the learned models. The learned linear models can fit the target data very well. This shows that our DARN can learn a meaningful model for a specific target domain, using only labelled source data and unlabelled target data. Second, Fig. 5c shows the source domain weights (the $\boldsymbol{\alpha}$) for each target domain after training. The weight colors correspond to the colors in Fig. 5a. It is noticeable that DARN can focus well on the respective relevant source domains for each target domain and ignore the rest. Note that these values are automatically learned during the training of the model.

## D    MODEL ARCHITECTURE FOR DIGIT RECOGNITION

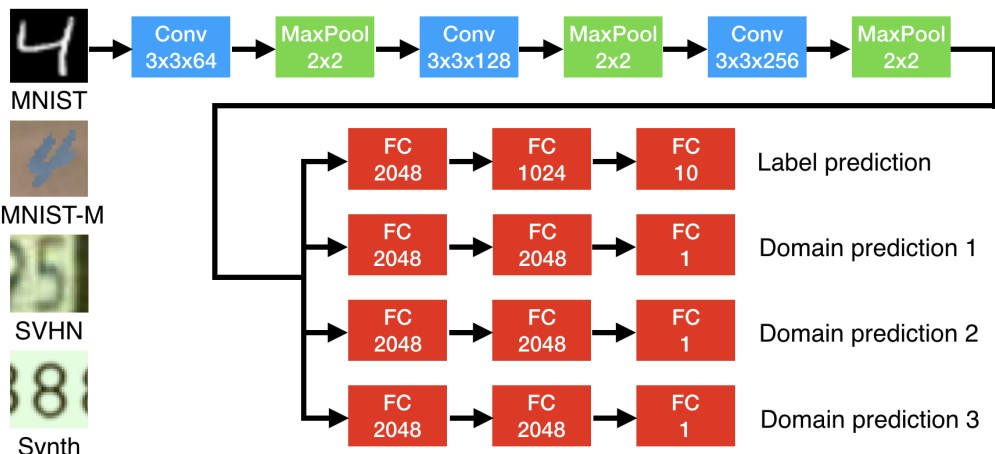

Figure 6: Model architecture for the digit recognition.

