# OpenReview forum: "Domain Aggregation Networks for Multi-Source Domain Adaptation"
_ICLR.cc/2020/Conference — Reject_

### Official Review · AnonReviewer1 · 2019-10-11
**Official Blind Review #1**

**Rating:** 3

**Review:**



###Summary###
This paper tackles the multi-source domain adaptation by aggregate multiple source domains dynamically during the training phase. The observation is that in many real-world applications, we want to exploit multiple source datasets of similar tasks to learn a model for a different but related target datasets.

Firstly, the paper derives a multiple-source domain adaptation upper-bound from single-to-single domain adaptation generalization bound, based on the theoretical work from Cortes et al (2019). The idea is similar to Zhao et al (2019), which introduces a weighted parameter \alpha to combine the source domains together.

Secondly, based on the theoretical result, the paper proposes an algorithm to minimize the upper bound of the theoretical result. The upper bound can be simplified as the quartic form (Eq. 4) and can be optimized with the Lagrangian form. Since no closed-form expression for the optimal v can be derived, the authors propose to use binary search to find it.

Based on the theoretical results and the algorithm, the paper introduces Domain AggRegation Network (DARN), which contains a base network for feature extraction, h_y to minimize the task loss and h_d to evaluate the discrepancy between each source domain and target domain.  The loss is aggregation with the parameter \alpha.

Finally, the paper conduct experiments on sentimental analysis benchmark, Amazon Review and digit datasets. The paper selects MDAN, DANN, MDMN as the baselines. On the amazon review dataset, the performance of the proposed DARN model is comparable with the MDMN baseline. On the digit dataset, the model can outperform the baselines.


### Novelty ###

The theoretical results in this paper are extended from Cortes et al (2019) and Zhao et al (2018).  Thus, the theoretical contribution of this paper is limited.

The algorithm proposed in this paper is interesting. However, the motivation of the proposed method is to minimize the upper bound, not the loss itself, i.e. L_T(h, f_T). Intuitively, when the upper bound of the loss is minimized, it will be beneficial to minimize the loss itself. But it's not guaranteed as the upper bound contains other variables, such as the number of training samples and model complexity. If the training samples and model complexity (think about the parameters in the deep models) are significantly large, the upper bound of the loss might be also very large.

As for the experimental results, the paper only provides results on the sentimental analysis results and digit datasets, which are small benchmarks. The selected baselines are not sufficient. The improvement from the baselines is also limited.



###Clarity###

Overall, the paper is well organized and logically clear. The images are well-presented and well-explained by the captions and the text.

The derivation of the algorithm in Sec 3.2 is logically clear and easy to follow.

###Pros###

1) The paper proposes a new theoretical upper-bound based on the prior works, the upper-bound and its derivation are interesting and heuristic to the domain adaptation research community.
2) The paper is applicable to many practical scenarios since the data from the real-world application is typically collected from multiple sources.
3) The paper is overall well-organized and well-written. The claims of the paper are verified by the experimental results.

###Cons###

1) The critical issue of this paper is that the algorithm is designed to minimize the upper bound. The idea is intuitive when the upper bound is small. However, the proposed upper bound in the paper involves other parameters, such as the model complexity and the number of training samples.
It's an intuitive idea to weight different source domains in multi-source domain adaptation. The paper derives the weight by the Lagrangian form to minimize the upper bound. While another trivial trick is to evaluate \alpha by the domain closeness between each source domain with the target domain.
2) The experimental results provided in this paper are weak. In the abstract and introduction,  the paper motivates the multi-source domain adaptation (MSDA) problem by arguing that the MSDA has a lot of real applications.  But the paper only provides empirical results on sentimental analysis and digit recognition.  Besides, the results on the sentimental analysis are comparable with the compared baselines.

It will be also interesting to see how does the proposed method perform on large-scale datasets such as DomainNet and Office-Home dataset:
DomainNet: Moment Matching for Multi-Source Domain Adaptation, ICCV 2019. http://ai.bu.edu/DomainNet/
Office-Home: Deep Hashing Network for Unsupervised Domain Adaptation, CVPR 2017. http://hemanthdv.org/OfficeHome-Dataset/

3) The novelty of this paper is incremental as the theoretical results are extended from Cortes et al (2019) and Zhao et al (2018).

Based on the summary, cons, and pros, the current rating I am giving now is weak reject. I would like to discuss the final rating with other reviewers, ACs.
To improve the rating, the author should explain the following questions:
1). Why minimizing the upper bound in this scenario would help to minimize the loss on the target domain, when the upper bound can be substantially huge? How about just evaluate \alpha by the closeness of the source domain with the target domain?
2). In the introduction, the paper motivates the multi-source domain adaptation (MSDA) problem by arguing that the MSDA has a lot of real applications. While the experiments are only performed on sentimental analysis and digit recognition. How about evaluating the proposed methods on real image recognition such as DomainNet or Office-Home?



**Experience Assessment:**

I have published one or two papers in this area.

**Review Assessment: Checking Correctness Of Derivations And Theory:**

I assessed the sensibility of the derivations and theory.

**Review Assessment: Checking Correctness Of Experiments:**

I carefully checked the experiments.

**Review Assessment: Thoroughness In Paper Reading:**

I read the paper thoroughly.

---

> ### Author Response · Authors · 2019-11-12
> **Additional Explanation and Results**
>
> Thank you for your thorough assessment and helpful comments! To answer your two questions:
>
> 1) Upper bound
> You are right that it would be ideal to optimize the target loss L_T(h, f_T) directly. However, this is not possible because we do not have labelled target data (i.e., f_T is unknown).  Minimizing an upper bound is arguably the only viable option *with theoretical generalization guarantee*. It is a common practice in the domain adaptation community (Mansour et al., 2009a, 2009b; Ben-David et al. 2007, 2010; Cortes and Mohri, 2011), and it is essentially the key idea of PAC learning and generalization analysis (Györfi et al., 2006; Schölkopf et al., 2002; Vapnik, 2013). Besides, our method *directly* optimizes the upper bound without resorting to heuristics, unlike prior methods.
>
> Ref:
> - Györfi, L., Kohler, M., Krzyzak, A. and Walk, H., 2006. A distribution-free theory of nonparametric regression. Springer Science & Business Media.
> - Schölkopf, B., Smola, A.J. and Bach, F., 2002. Learning with kernels: support vector machines, regularization, optimization, and beyond. MIT press.
> - Vapnik, V., 2013. The nature of statistical learning theory. Springer science & business media.
>
> 2) More experiment
> Thank you for pointing out these datasets. We have conducted further experiments on the Office-Home dataset,  using the ResNet50 as the backbone architecture and changing the classification head to 65 classes. As we can see in the new Section 5.3, our method achieve state-of-the-art performance and outperform all alternatives; these results are statistically significant.
>
> In addition, we have added one more competitive baseline (M3SDA) from the DomainNet paper you mentioned, using their public code with a few necessary adjustments for each dataset (e.g., network architecture, etc). We ensure that all methods use the same backbone architecture for a fair comparison. Again, our method outperform M3SDA in all datasets.
>
> If you have any other comments or concerns, we are happy to provide further feedback. Thank you!

---

### Official Review · AnonReviewer2 · 2019-10-27
**Official Blind Review #2**

**Rating:** 6

**Review:**

This paper studies multi-source domain adaptation problem. First this paper proposes a new theory for this domain that extends generalized discrepancy theory to multi-source setting. After derive a new generalization bound, this paper also proposes a new method based on the theory. Evaluation on real world datasets are proposed to show the efficiency of the proposed method.

+ The theory in this paper improve bounds for multi-source DA in previous paper. The new bound provides new insight and helps the design of algorithm.
+ This paper proposes elegant method to tackle new terms in the loss function and gives its complexity analysis.
+ The evaluation results show that the algorithm is efficient.

Comments:
- The main contribution of the proposed theory is the alpha term. Is \eta_{\mathcal{H}} is a better estimation that previous adaptability term in multi-source DA? What is the role of \eta_{\mathcal{H}} in the algorithm design? How to control it in empirical algorithm?
- The evaluation of the proposed method is not complete. Some baseline DA methods [A, B] and datasets [C, D] are not considered.

[A] S. Sankaranarayanan, Y. Balaji, C. D. Castillo, and R. Chellappa. Generate to adapt: Aligning domains using generative adversarial networks. In The IEEE Conference on Computer Vision and Pattern Recognition (CVPR), June 2018.
[B] Saito, Kuniaki, et al. "Maximum classifier discrepancy for unsupervised domain adaptation." Proceedings of the IEEE Conference on Computer Vision and Pattern Recognition. 2018.
[C] K. Saenko, B. Kulis, M. Fritz, and T. Darrell. Adapting visual category models to new domains. In European Conference on Computer Vision (ECCV), 2010.
[D]H.Venkateswara, J.Eusebio, S.Chakraborty, and S.Panchanathan. Deep hashing network for unsupervised domain adaptation. In IEEE Conference on Computer Vision and Pattern Recognition (CVPR), 2017.


**Experience Assessment:**

I have published one or two papers in this area.

**Review Assessment: Checking Correctness Of Derivations And Theory:**

I carefully checked the derivations and theory.

**Review Assessment: Checking Correctness Of Experiments:**

I carefully checked the experiments.

**Review Assessment: Thoroughness In Paper Reading:**

I read the paper thoroughly.

---

> ### Author Response · Authors · 2019-11-12
> **Additional Explanation and Results**
>
> Thank you for your feedback and additional references! To address the comments:
>
> - As we explained in Thm.1, \eta_{\mathcal{H}} is a constant measuring how well the model family \mathcal{H} can fit the true models from both domains. Estimating this term requires *labelled* target samples, which is usually unavailable in domain adaptation. However, when we have access to a handful of labelled target data, we can certainly estimate this term and perform model selection (e.g., choosing neural network models \mathcal{H}) better, meaning that we can find better values for alphas, and so achieve even better adaptation performance.
>
> - Yes, there are many methods that conduct single-source to single-target adaption in the literature. However, our main focus is *multi-source* to single-target adaptation. This is why our comparisons focus on similar methods, that also use multiple sources at the same time. They are generally more competitive than single-source methods. Following Reviewer #1's suggestions, we added one additional state-of-the-art competitor, Moment Matching for Multi-Source Domain Adaptation (M3SDA) (Peng et al., 2019), to our experiments. Moreover, we also add the challenging Office-Home dataset as you suggested[D]; results can be found in the new Section 5.3. The results with the new competitor, and on both the earlier datasets and the new one, show that our method outperforms the competition, especially on the Office-Home dataset, in which we achieve state-of-the-art performance.
>
> If you have any comments or concerns, feel free to leave a message here and we can discuss further. Thank you!

---

### Official Review · AnonReviewer4 · 2019-11-06
**Official Blind Review #4**

**Rating:** 6

**Review:**

This paper proposed an aggregation algorithm (DARN) for the multi-source domain adaptation problem which is highly useful in real-world applications.
The proposed method is based on the theoretical extension of the single-source domain discrepancy measure proposed by Mansour et al. 2009 to the multi-source setting.
This paper also showed the effectiveness of the proposed method on some real-world datasets.

Strengths
The paper introduces new technical insights to understand their bound, e.g. effective sample size.
The paper proposed the way to estimate coefficient, optimal \alpha, with theoretical justification, and I think this is the biggest contribution of this paper and is interesting.
The proposed method is also able to be used in the regression task since it is based on the disc which can be estimated in the regression task.

Weakness
The main theorem of this paper is an extension of existing methods, so the novelty of theoretical analysis is somewhat limited.
A naive approach to estimate coefficient with single-source domain discrepancy measures such as [1]Mansour (2009), [2,3] Ben-David(2007, 2010), [4] Kuroki et al (2019), and W1-distance is not considered.
Experimental results itself are fine but not complete.
  - Although disc can be easily estimated in the regression task (differently from d_A distance which is a special case of disc), there are no experimental results of the regression task even in the synthetic data.
  - It would be also better to show the coefficient of existing methods that have no theoretical justification.
  - It would be better to compare with a naive approach that uses domain discrepancy between each source and target as (fixed) coefficient since this approach such as Mansour (2009), Ben-David(2007, 2010) and Kuroki et al (2019) which explicitly consider the hypothesis class has theoretical justification in the form of generalization error bound in the target domain.

Overall, I like the approach of the proposed method, especially tuning coefficient during training procedure although novelty in the theoretical analysis is somewhat limited.
So this work has to be supported with more detailed experimental results to express the potential of this approach fully.
For this reason, I think it is okay but not good enough at this time.

****After the authors' response****
Increase rating.

[1] Yishay Mansour, Mehryar Mohri, and Afshin Rostamizadeh. Domain adaptation: Learning bounds and
algorithms. In COLT, 2009.
[2] Shai Ben-David, John Blitzer, Koby Crammer, and Fernando Pereira. Analysis of representations for
domain adaptation. In NeurIPS, 2007.
[3] Shai Ben-David, John Blitzer, Koby Crammer, Alex Kulesza, Fernando Pereira, and Jennifer Wortman
Vaughan. A theory of learning from different domains. Machine Learning, 2010.
[4] Seiichi Kuroki, Nontawat Charoenphakdee, Han Bao, Junya Honda, Issei Sato, and Masashi Sugiyama.
Unsupervised domain adaptation based on source-guided discrepancy. In AAAI, 2019.

**Experience Assessment:**

I have published one or two papers in this area.

**Review Assessment: Checking Correctness Of Derivations And Theory:**

I carefully checked the derivations and theory.

**Review Assessment: Checking Correctness Of Experiments:**

I assessed the sensibility of the experiments.

**Review Assessment: Thoroughness In Paper Reading:**

I made a quick assessment of this paper.

---

> ### Author Response · Authors · 2019-11-12
> **Additional Explanation and Results**
>
> Thank you for your insightful feedback and suggestions! To address your comments:
>
> - Following your suggestion, we have added one experiment on a synthetic regression task in Appendix C. Here, we show that our method can learn meaningful models for target domains, and also learn the source domain weights in a way that selects only relevant source domains for training. More importantly, we can learn the model and the domain weights simultaneously, unlike many existing works that use two-stage learning (learn the weights then the model).
>
> - We also followed your suggestion and show the domain weights of MDMN on the Amazon dataset for comparison (see the new Section 5.4). As you mentioned, our weights are theoretically justified while theirs are only heuristically computed. We see that the weights provided by MDMN are not very stable, changing from one source domain to another drastically during training. This instability makes their weights difficult to interpret.
>
> - W.r.t. the naive method of using (fixed) coefficients. Note that whichever domain discrepancy is used, it has to depend on the data representation. For complicated models like deep neural networks, the feature representations (hidden layers) are changing constantly during training and arguably there is no "right" way to compute *fixed* coefficients/weights based on ever-changing representations. Computing the coefficients directly from the images is extremely difficult, if not impossible, because calculating the domain discrepancy using such high-dimensional data is not feasible.
> Note that MDMN DOES use W1-distance to compute domain weights, and our comparison in Section 5.4 shows that it is not very stable, as mentioned above. To summarize, there is no easy way to compute meaningful fixed coefficients and our method is indeed suitable for dynamic representations during neural network training.
>
> We hope our explanation and additional results address and resolve your concerns. If you have any other comments, we are happy to have further discussions. Thank you!

---

> > ### Comment · AnonReviewer4 · 2019-11-13
> > **Response to authors**
> >
> > Thank you for your careful response and additional experimental results!
> > Most of my concerns are solved in the revised paper and by the authors' response.
> >
> > So, I decided to increase my score, but I also have to consider the situation that I make a quick assessment.
> >
> > Comments:
> > Obviously, \eta_{\mathcal{H}} is impossible to estimate in the unsupervised domain adaptation.
> > So many researches assume that it is small, and also in this work.
> > Since there might exist some concerns on this term, I think it is better to add research by [1] Ben-David(2010b) which showed that the assumption of small \eta_{\mathcal{H} is necessary for domain adaptation to succeed as a reference.
> > It means that we cannot guarantee the success of domain adaptation algorithms always "only with small discrepancy measure".
> > However, I believe that it is quite natural to assume \eta_{\mathcal{H} is small in real-world applications and that is why we use domain adaptation.
> > In other words, if we do not have any prior knowledge between the source and the target domains, we might not try to adapt using specific domains as a source.)
> >
> > [1] Shai Ben-David, Tyler Lu, Teresa Luu, and Dávid Pál. Impossibility theorems for domain adaptation.
> > In AISTATS, 2010b.

---

> > > ### Author Response · Authors · 2019-11-15
> > > **Thank you**
> > >
> > > Thank you for your additional comments! We will include and discuss the reference in the revised version.

---

### Author Response · Authors · 2019-11-12
**Revision**

We thank the reviewers for the constructive feedback and insightful comments. We revised our paper accordingly with additional results. The major revision includes:

1. We added the Office-Home dataset and compared our method to the competing alternatives in Section 5.3. Our method achieves state-of-the-art performance on this multi-source adaptation problem.

2. We added one more competing method (Moment Matching for Multi-Source Domain Adaptation, M3SDA for short) from the literature in all three experiments. We use the original authors' implementation with a few necessary adjustments (using the same neural network architecture for a fair comparison, modify the classification head to according to the number of classes, etc). We find that our method outperforms M3SDA over all three experiments.

3. In Section 5.4, we compare the domain weights of our method and MDMN. We can see that the weights of MDMN are less stable and hard to interpret.

4. Appendix C now demonstrates how our method works for a synthetic regression problem. Our method can simultaneously learn meaningful models for the target domains, and the domain weights, to focus on relevant source domains during training.

Please also see the individual feedback we provided for each reviewer.

---

### Decision · Program_Chairs · 2019-12-19

**Decision:**

Reject

**Comment:**

Thanks for the detailed replies to the reviewers.
Their score was slightly improved, this paper is still below the bar given high competition of ICLR2020.
For this reason, we decided not to accept this paper.